# The spatial organization of intra-tumour heterogeneity and evolutionary trajectories of metastases in hepatocellular carcinoma

Weiwei Zhai[1],[*],[**], Tony Kiat-Hon Lim[2],[*], Tong Zhang[1],[*], Su-Ting Phang[3], Zenia Tiang[1], Peiyong Guan[1], Ming-Hwee Ng[1],[4], Jia Qi Lim[1],[4], Fei Yao[1], Zheng Li[1], Poh Yong Ng[1], Jie Yan[1], Brian K. Goh[5], Alexander Yaw-Fui Chung[5], Su-Pin Choo[6], Chiea Chuen Khor[1], Wendy Wei-Jia Soon[1], Ken Wing-Kin Sung[1],[7], Roger Sik-Yin Foo[1],[8],[**] & Pierce Kah-Hoe Chow[3],[5],[9],[**]

Hepatocellular carcinoma (HCC) has one of the poorest survival rates among cancers. Using multi-regional sampling of nine resected HCC with different aetiologies, here we construct phylogenetic relationships of these sectors, showing diverse levels of genetic sharing, spanning early to late diversification. Unlike the variegated pattern found in colorectal cancers, a large proportion of HCC display a clear isolation-by-distance pattern where spatially closer sectors are genetically more similar. Two resected intra-hepatic metastases showed genetic divergence occurring before and after primary tumour diversification, respectively. Metastatic tumours had much higher variability than their primary tumours, suggesting that intra-hepatic metastasis is accompanied by rapid diversification at the distant location. The presence of co-existing mutations offers the possibility of drug repositioning for HCC treatment. Taken together, these insights into intra-tumour heterogeneity allow for a comprehensive understanding of the evolutionary trajectories of HCC and suggest novel avenues for personalized therapy.

[1] Department of Human Genetics, Genome Institute of Singapore, Agency for Science, Technology and Research, 60 Biopolis Street, Genome, #02-01, Singapore 138672, Singapore. [2] Department of Pathology, Singapore General Hospital, Singapore 169608, Singapore. [3] Division of Surgical Oncology, National Cancer Centre, Singapore 169610, Singapore. [4] School of Biological Sciences, Nanyang Technological University, Singapore 637551, Singapore. [5] Department of Hepato-Pancreato-Biliary and Transplant Surgery, Singapore General Hospital, Singapore 169608, Singapore. [6] Division of Medical Oncology, National Cancer Centre, Singapore 169610, Singapore. [7] School of Computing, National University of Singapore, 13 Computing Drive, Singapore 117417, Singapore. [8] Cardiovascular Research Institute, National University of Singapore, National University Healthcare System, Singapore 119228, Singapore. [9] Office of Clinical Sciences, Duke-NUS Graduate Medical School, Singapore 169857, Singapore. * These authors contributed equally to this work. ** These authors jointly supervised this work. Correspondence and requests for materials should be addressed to W.Z. (email: zhaiww1@gis.a-star.edu.sg) or to R.S.-Y.F. (email: foosyr@gis.a-star.edu.sg) or to P.K-H.C. (email: pierce.chow.k.h@singhealth.com.sg).

Liver cancer is the fifth most common cancer globally but the second most important cause of cancer deaths[1]. An important contribution to this grave prognosis is the paucity of efficacious systemic treatment[2]. The current systemic therapy sorafenib has no validated biomarker and improves survival in advanced hepatocellular carcinoma (HCC) by only a few months (2.8 months in Caucasians[3] and 2.3 months in Asians[4]).

Understanding the origin and development of intra-tumour heterogeneity (ITH) has the potential to yield important insights for therapeutic target selection and drug development[5]. Although ITH has been observed for many clinical and molecular phenotypes in HCC (for example, histologic grade[6], cell proliferation[7], morphological and immunohistochemical profiles[8]), ITH survey at the genomic level for single lesion HCC has only been explored using single patient cases[9–11], revealing appreciable variability possibly driven by non-Darwinian forces[9]. A related study focusing on multi-lesion Hepatitis B Virus (HBV)-positive HCC discovered variable extent of genetic variability[12]. Considering the diverse aetiologies and ethnic backgrounds contributing to HCC[2], a systematic survey of ITH across diverse patient groups is fundamental to understanding the evolutionary trajectories of HCC.

In this study, using multi-regional sampling of nine resected HCC with different aetiologies, we find that the phylogenetic relationships of tumour sectors from the same patients have diverse levels of genetic sharing, spanning early to late diversification. Distinct to what is found in colorectal cancers where genetic variegation accompanies the development from adenoma to carcinoma[13], a large proportion of HCC displays a clear geographic segregation where spatially closer sectors are genetically more similar. Multi-sector analysis of the two intra-hepatic metastases revealed much higher genetic variability than their primary tumours, suggesting that intra-hepatic metastasis is accompanied by rapid diversification at the distant location. Our results draw a comprehensive understanding of HCC development and pinpoint unique evolutionary trajectories for intra-hepatic metastases.

## Results

**Spatial sectoring and sequencing.** We studied early pathological stages (American Joint Committee on Cancer stages I and II (ref. 14) of surgically resected HCC with clear microscopic margins (R0 resection) from patients with good liver function (Child-Pugh A). Eight HCC were solitary and one had a concomitant second smaller tumour in the contralateral lobe, treated by intra-operative radiofrequency ablation. These HCC have arisen from chronic viral hepatitis B (five cases), chronic viral hepatitis C (three cases) and diabetes mellitus with steato-hepatitis (one case) from five ethnic groups namely Singapore Chinese, Singapore Malay, Indonesian, Cambodian and Burmese (Supplementary Table1).

To elucidate the spatial organization of the ITH, we sampled a grid of sectors through a central slice of each tumour (Fig. 1a). Forty-seven whole-genome (WGS, 37 × coverage) and 13 whole-exome sequencing (WES, 85 × coverage) were performed across the 9 tumours (Fig. 1a and Supplementary Table 2). Two of the nine patients were subsequently diagnosed with intra-hepatic metastases and were both surgically resected at approximately 12 months after the index surgery (interval of 384 days in patient 1 and 381 days in patient 2). These resected metastases were similarly subjected to multi-sectoring (two sectors for the first patient and four sectors for the second patient).

Raw sequence data were processed through a series of cancer genomic analyses pipeline (Methods). Through experimental validation, we found that the true positive rate of the somatic variants was high (96%, Supplementary Note 1 and Supplementary Data 1). In all patients, the rates of somatic mutation in the primary tumour ranged from 2.95 to 17.23 single-nucleotide variants per Mb (Fig. 1b) and were intermediate compared with many other common cancer types (Supplementary Fig. 1 and Supplementary Note 2). Interestingly, patient 9 who had much higher mutation burden than other patients carried a strong AA signature possibly driven by herbal chemicals (Supplementary Fig. 2)[15,16]. Using genome-wide copy number and minor allele frequency profiles, we estimated the tumour purity across the samples (Methods and Fig. 1b). All the sectors in our sample showed good tumour purity (mostly between 0.5 and 0.9) with a mean purity of 0.69. This matched earlier observations that tumour cell content in HCC is generally much higher than many other cancer types[17].

**Driver mutations and copy number variations across cases.** By extracting a total of 1,185 WGS/WES of liver cancer genomes from open sources together with the 9 cases here (Methods), we identified 65 significantly mutated driver genes for HCC[18] (Supplementary Data 2). The most frequent driver genes for HCC were *CTNNB1* and *TP53* (Fig. 1b). Of these 65 driver genes, 16 genes bore at least 1 non-synonymous mutation in the 9 patients (Fig. 1b). *TP53* was the most frequent driver with hotspot mutations identified in six patients. Inspection of the location of these mutations revealed that patient 3 and patient 6 shared *TP53* mutations at the same location but with different alternative alleles (Supplementary Table 3). Combining these with previously identified *TERT* promoter mutations, most of the somatic mutations in the driver genes were in higher allele frequencies than the background mutations ($\chi^2$-test, P-value $<1E-6$) and were shared among all tumour sectors (Fig. 1b). This suggested that known driver mutations arose very early in the tumourigenesis of HCC[19].

Similar to point mutation analyses, we also surveyed copy number variations (CNVs) across the genome for all the samples (Supplementary Fig. 3)[20]. The most significantly mutated CNVs were amplifications in chromosome 1q, 6p and 8q, as well as deletions in 4q, 8p, 13q and 17p (Methods). Chromosome 8q amplification was observed for patients 1, 5 and 6, and chromosome 4q deletion in patients 1, 2, 3, 4 and 7 (Fig. 1c and Supplementary Fig. 4). Inspecting previously identified HCC drivers as well as genes in the cancer gene census within these segments, important cancer driver genes such as *CCND1* were found to be amplified in patient 6 and patient 9 (Fig. 1d). In general, the CNV profiles across the genome were quite similar across sectors in HCC[12]. This is consistent with the notion that genome instabilities are early events in tumourigenesis[21–24].

**The spatial organization of genetic variability.** Using somatic variants called from individual sectors, we constructed the evolutionary relationships for primary tumour sectors (Methods). Mutations that were common to all sectors appeared on the trunk of the tree and mutations that were private to subsets of the samples appeared on branches. Among all cases, the level of differences between tumour sectors varied considerably (Fig. 2). To compare genetic diversity across cases, we plotted the relationship between detected variability and the number of samples (Methods). Using the amount of mutation detected using single samples as the baseline, we found that observed variability increases rapidly with the multi-regional sampling of tumour sectors (Fig. 3a). Using the slope of the upward trajectories as a surrogate for the degree of ITH in a tumour, the wide variation in the slope suggested that the evolutionary trajectories of HCC were very diverse (Fig. 2).

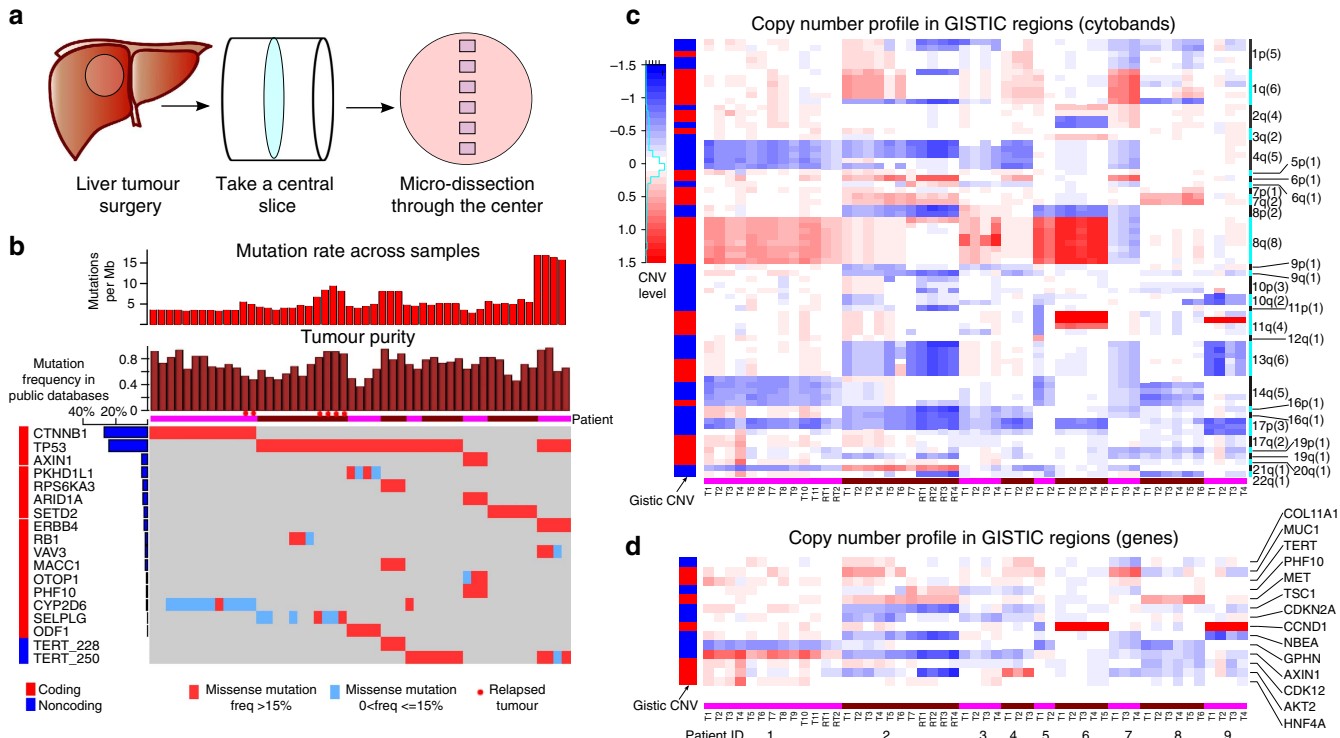

**Figure 1 | Spatial sampling and genomic profiles. (a)** A schematic flow of our sectoring design. A central slice was cut from the patient tumour. A linear grid of tumour sectors was then harvested. **(b)** Oncoprint plot for 18 HCC drivers across 9 patients. Mutation rates, tumour purity and mutation presence data are shown in the top, middle and bottom panels. The mutation frequencies of each HCC driver gene (left side bar) were extracted from a large collection of public data sets (Methods). Sectors from each patient are ordered from left to right according to their names (T1 being the most left sector). Red shows the mutations whose frequencies are >15% and blues are those with frequencies <15%. For patient 3, we have no information for TERT promoter mutations due to exome sequencing. **(c)** Copy number profiles at GISTIC cytobands. Each cytoband is one row and chromosomal arms of the cytobands are shown on the right. The number in the parentheses is the total number of cytobands in that chromosome arm. Precise cytoband IDs are listed in Supplementary Fig. 4. Left side bar is the copy number profile in the GISTIC analysis. Red designates amplifications and blue is for deletions. **(d)** Copy number profiles of potential driver genes. Format is as **c**.

Across nine cases, the phylogenetic trees took a variety of forms. For example, patients 1, 4 and 6 showed typical star genealogies with very short internal branches, indicating rapid population expansion in the history of the tumour[25]. Using a computational procedure from Statistical Phylogenetics known as bootstrapping[26], we assessed the confidence in the topological relationships among the sectors. Statistical analyses showed that the evolutionary relationships are highly consistent (bootstrap score >0.7) with the exception of patients 1 and 4, where the population diversification was very recent and the sectors were highly similar to each other (Supplementary Fig. 5). WGS provided high-resolution for the phylogenetic relationships across all the other patients including patient 6.

Across all phylogenetic trees with sufficient divergence (that is, except patients 1 and 4), sectors from one end of the tumour consistently grouped together, with separation of the two clades setting apart from the tumour centre (Fig. 2). This suggested that, HCC generally arose from ancestral clones found in the centre of the tumour and genetic lineages diverged as the tumour grew outwards. When the ancestral clone at the centre of the tumour was sampled in the tumour sectoring, the phylogenetic tree took a slightly different form where the ancestral clone (that is, basal lineage) would first branch off (Fig. 2 legend). This was observed in patients 6 and 7 (that is, T2 for patient 6 and T3 for patient 7 in Fig. 2).

The phylogenetic analysis was based on the presence and absence of somatic changes across tumour sectors, which only utilized a portion of the available genetic information. Next,

we quantified the genetic relationships among populations by measuring allele frequency differences among samples using a classical metric that was derived from population genetics (defined as the Fixation Index or Fst[27]). Fst measures the proportion of the total variance in allele frequency caused by frequency differences between populations. When we plotted the levels of population differentiation as a function of the physical distances between tumour sectors, controlling for tumour purity (Methods and Supplementary Note 3), a clear pattern was observed. Tumour sectors that are physically closer tend to be genetically more similar (defined as isolation-by-distance (IBD) relationship in Evolutionary Genetics[28]) (Fig. 3b and Supplementary Fig. 6). Patient with late diversification (for example, patients 1 and 4) show much weaker isolation with mild slope in the linear regression, matching the pattern found in the phylogenetic inference (Fig. 2 and Supplementary Fig. 6).

**The origin of IBD pattern and spatial modelling.** The IBD pattern together with the phylogenetic relationship strongly implied a range expansion dynamics in the growth of HCC[29]. To test this hypothesis, we built a spatial model to simulate tumour growth and outward expansion of cell populations (Fig. 3c and Methods)[13,30]. We found that mutations arose in the history of tumour development tend to segregate in clear geographic locations (Supplementary Fig. 7)[13,29,30]. By sampling different tumour sectors and measuring their genetic differentiation (Fst), we found that a variety of growth models can contribute to the

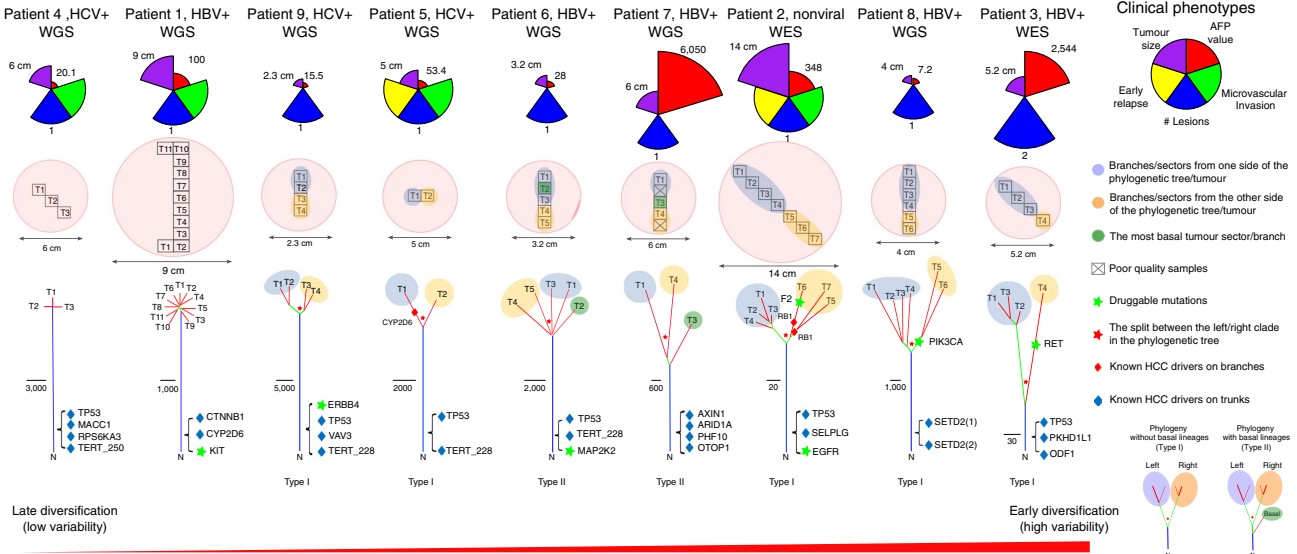

**Figure 2 | The phylogenetic relationship and clinical phenotypes.** The spatial relationship in tumour sectoring (shown in the red pie chart); the phylogenetic tree relating to different sectors for each of the nine cases. For patients with enough of genetic heterogeneity (that is, except patient 1 and 4), blue and orange eclipses mark different genetic lineages and their physical locations in the tumour. Patients are ranked by their level of intra-tumour heterogeneity from left to right. The splits between the left and right lineages/sectors are marked with a red star. In patient 6 and 7, the most basal genetic lineages were detected and were labelled as green circles. Patient clinical information was displayed with coxcomb plots. In patient 7, two sectors were dropped due to poor sample quality. Two different mutations in SETD2 (chr3 47162099 and 47161730) happened in patient 8. Two different tree topologies are found across nine patients. If the most basal clone is sampled, the basal lineage will first branch off followed by the sectors from two sides of the tumour (type II). Otherwise, the phylogenetic tree will just consist of two deeply separated clades (type I).

observed IBD pattern observed in HCC (Supplementary Note 4). Interestingly, when mimicking a sampling strategy similar to our sequencing approach (Fig. 3d), we could easily recapitulate our observation in the real data (for example, bipartite distribution of genetic lineages across the tumour centre and the IBD pattern). Overall, we conclude that our data are compatible with the model of a range expansion process[13,29,30] driven by an ancestral clone from the centre of the tumour.

**Clinical phenotypes and targetable mutations**. Previous studies found that tumours with higher levels of genetic diversity (that is, early diversification) tend to result in poorer clinical prognosis across several cancer types[31,32]. We inspected clinical phenotypes across these nine patients and correlated them with their levels of ITH (Fig. 2). Even though there was a tentative trend of higher alpha-feto protein in tumours with higher ITH, most of the other tumour characteristics did not correlate with the evolutionary trajectories in our collection. Quite possibly, a larger cohort study will be needed to explore the potential link between clinical phenotypes and ITH for HCC[33].

Although there are currently no established therapies against previously described HCC drivers, many co-existing mutations (for example, *EGFR* and *KIT*) can be targeted with existing therapeutics (Methods and Fig. 2). Although the locations of most of these mutations were not necessarily found on the exact same genomic loci of well-known oncogenes (for example, *EGFR* L858R for lung cancer), several of them were on important functional domains (Supplementary Table 4). Given that many cancer drugs target important functional pathways rather than specific mutations, genetic changes detected through ITH analyses in HCC may probably allow for the re-positioning of existing therapeutics through better patient stratification[34].

**High variability in the metastatic tumour**. From patients 1 and 2, we were able to obtain multiple sectors from resected intra-hepatic metastases. The monophyletic relationships (forming a single clade) seen in the sectors from metastatic tumours suggested a single point of origin for the metastases (Fig. 4a). Interestingly, in both patients, the topological relationship between the metastatic clone and the primary tumour took two different forms. For patient 1, the metastatic clone connected to the trunk of the primary phylogeny and the genetic divergence occurred before the genetic diversification of the primary tumour. The opposite was true for patient 2. Lineage neighbouring to the T7 (the most outer tumour sector; Fig. 2) gave rise to the metastatic clone and the migration occurred after the diversification of the primary tumour. Intra-hepatic metastases in HCC could undertake two classical scenarios where metastatic clone can originate before (early migration) or after (late migration) diversification in the primary tumour.

During metastases, when a small group of cells (clone) leaves the parental population to form a newly established colony, two major expectations known as the founder effect[35] are generally anticipated. First, the migratory population will be quite different from the parental population due to the population bottleneck. Previous studies across several cancer types often reveal high genetic divergence between metastatic and primary tumour[36–39], matching the expectation from the founder effect. Second, the genetic variability will be much reduced in the new population. Surprisingly, neither of these two expectations was true in the metastases from both patients' tumours.

The divergence between the metastatic and primary tumour, represented by the migratory branch (Fig. 4a), was quite short in comparison with the branches of the tumour sectors from the primary tumour. The short migratory branches suggest that the metastatic clones did not further evolve from the primary tumour. In other words, following migration to the distant location, the migratory clone did not require further adaptation before the genetic diversification. This may be enabled by the local microenvironment of the regenerating liver following tumour resection[40]. Rapid growth stimulated by liver

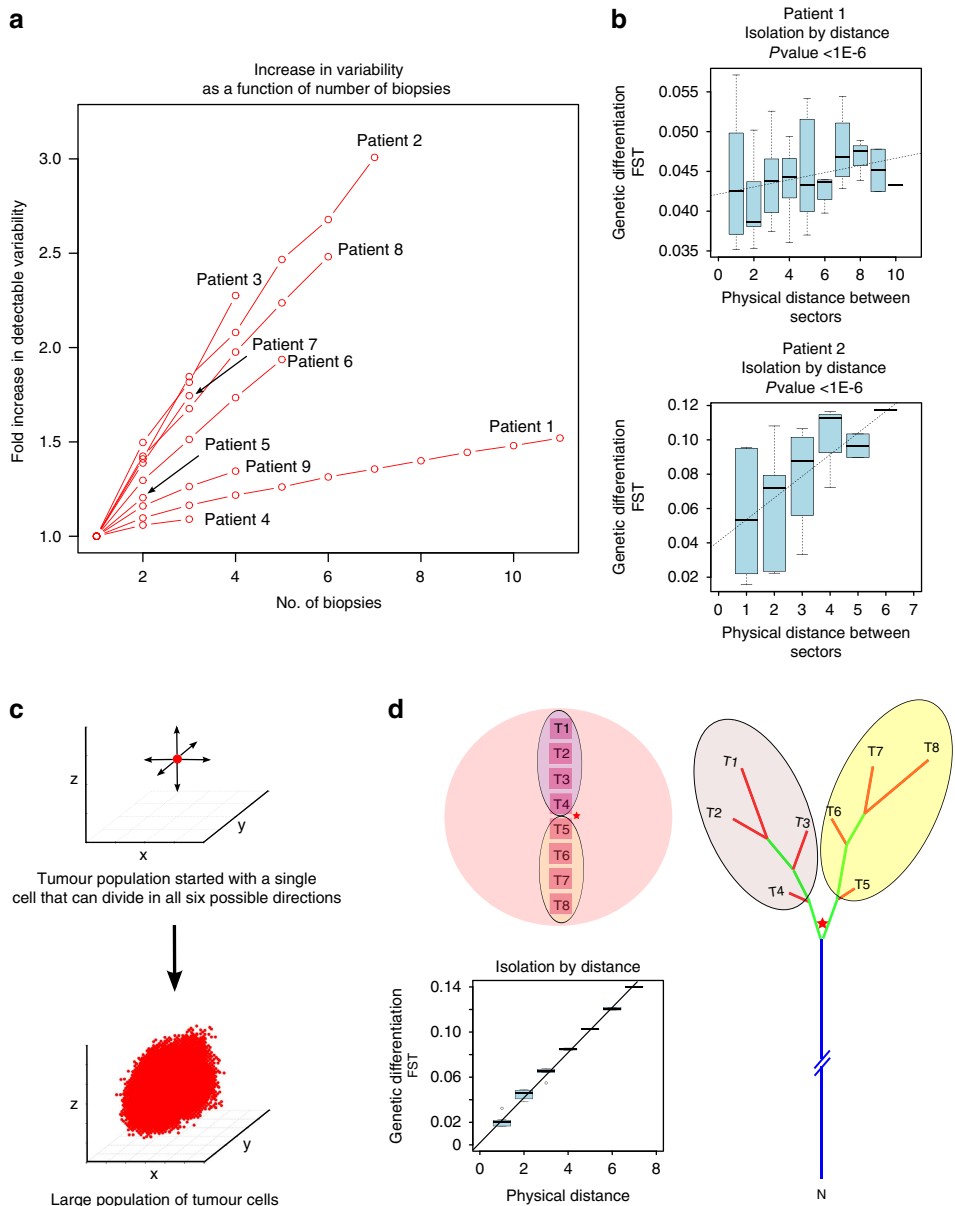

**Figure 3 | IBD pattern and the spatial modelling. (a)** The relationship between the number of biopsies and observed variability (see Methods). **(b)** IBD pattern for patient 1 and 2. The x axis is the physical distances between sectors and the y axis is the genetic differentiation (Fst) between the samples. Fst values from all sector pairs with the same physical distance were used to draw boxplots at each distance value. The regression line and the P-value are derived from the linear regression model (Methods). The boxplot plot is plotted using boxplot() with default settings from R (Methods). **(c)** Spatial modelling. Single cells were seeded in a three-dimensional grid and allowed to divide in all six directions until they reached a certain size. **(d)** Simulation results that match our data. The same sampling procedures were conducted on the simulated data. Phylogenetic trees and IBD pattern from the simulation were found to be similar to Figs 2 and 3b.

regeneration allows the metastatic tumour to quickly diversify and attain high genetic variability. The cases described above represent the first descriptions of the complete clinical trajectory of HCC (from primary resection to metastases) and such observations posit strong clinical implications on how systemic therapeutic intervention should be planned for the more effective control of HCC and the arising metastases (see Discussions).

**Dynamic viral integration across tumour evolution.** Another important driver of HCC development is the viral integration. Using a computational procedure to search for paired-end reads where one end maps to the human genome and the other end maps to the HBV genome, we identified viral integrations in four HBV-positive patients from our cohort[41] (patient 3 was omitted,

as it was not sequenced by WGS). HBV integrations were more abundant in patient 6 than in patients 1 and 8 (Fig. 4b, Supplementary Note 5 and Supplementary Table 5). Interestingly, no viral integration was found for patient 7, possibly due to low viral burden (Supplementary Table 6). In patients 1, 6 and 8, viral integrations were found in adjacent normal tissue, in addition to tumour sectors, indicating that HBV viral infection and integration were very active processes during HCC development and progression. When classifying the genome into integration hotspots (Methods and Supplementary Fig. 8)[41,42], two TERT gene integration and one non-coding integration hotspot on chromosome 10 (far from any known gene) were found[43]. The pattern of viral integration suggested that viral population evolved together with the tumour population and integration

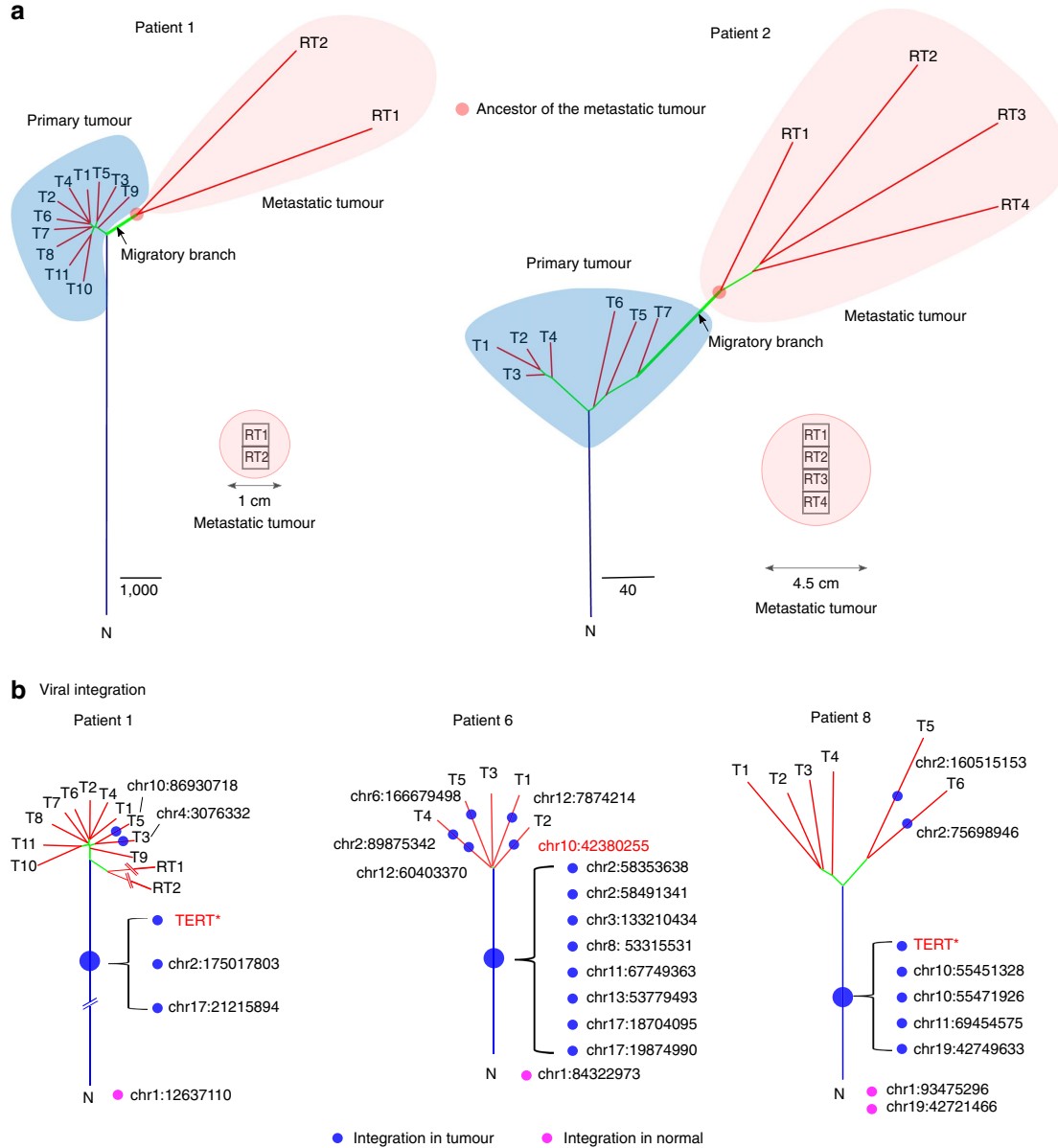

**Figure 4 | Intra-hepatic metastasis and viral integration. (a)** The phylogenetic relationship between metastatic tumour sectors and their primary tumour. The branch linking the ancestral metastatic clone to its primary is designated as the migratory branch. Red circles mark the ancestor of the metastatic tumour. **(b)** HBV integration in context of the phylogenetic relationship. Integration sites are shown on each individual lineage. Integrations in red are those in hotspots. Integrations on the trunk of trees are ordered by their genomic coordinates.

could be frequent in the history of tumourigenesis. Viral integration in the *TERT* gene occurred early in the history of tumour progression and was also found in all tumour samples for patients 1 and 8 (including the metastatic tumour for patient 1). Together, these results suggested that viral integration can be an important early driver for HCC[41] in patients with chronic hepatitis B.

## Discussions

In colorectal cancers, genetic lineages often segregate in a variegated pattern where cell populations from the left and right side of the tumour interleave in the phylogenetic relationship[13]. The transition from clear spatial organization to variegated pattern has been associated with the transformation from adenoma to carcinoma and the mixing of clones was suggested to be of the 'born to be bad' type[13]. In our analyses, HCC were largely organized spatially (including the metastatic tumour of patient 2) and showed a clear distribution compatible with range expansion dynamics[9]. Interestingly, computational modelling suggests that the patterns observed for HCC can be easily recapitulated using relatively simplistic models. This might be a distinctive feature of HCC that could be related to tissue anatomic organization (for example, the lack of mucosal crypts) or/and the liver regenerative capacity whereby the vast majority of hepatocytes appear to be self-renewed (as opposed to a stem cell hierarchy model) driving the rapid progression of HCC[44]. It remains to be tested whether other tumour types follow these disparate colon or liver growth patterns.

Our study offers the first insights into interesting dynamics in intra-hepatic HCC metastasis. The comparisons between metastatic and primary tumour have so far revealed high genetic divergence across several cancer types[36–39]. This is in stark contrast to our findings for HCC. If multi-sectoring of the

metastatic tumour had not been carried out, distant divergence would have been inferred from high ITH in the metastatic tumour[11,12]. The short migratory branch instead suggests minimal further adaptation (clonal sweeps) was required at the distant location. The higher genetic diversity specific to the metastatic tumour suggests that it is quite inadequate to use the primary tumour for therapeutic prediction.

The absence of further adaptation at distant metastasis could be a feature specific to the liver, where the organ microenvironment appears to be more homogeneous across the liver. Mechanisms allowing local migration in the primary tumour may overlap with pathways for intra-hepatic migration[45]. Intra-hepatic metastases therefore does not require drastic adaptations to survive growth and colonize. Another possible triggering factor for rapid metastatic outgrowth may be related to the intrinsic regenerative capacity of the liver. Upon partial hepatectomy, dramatic systemic and localized influences that promote normal hepatocyte regeneration is also likely to drive the growth of metastatic carcinoma cells within the organ[40].

The clear spatial segregation of genetic lineages in HCC is analogous to the annual ring in woody plants and they offer a useful approach for understanding clonal evolution in the context of cancer. As ancestral lineages tend to segregate inside the tumour, spatial sampling will allow the reconstruction of more realistic and clinically relevant pre-clinical models (for example, patient-derived cell lines) that reconstitute various levels of genetic transformation (some ancestral lineages and some derived/evolved). Such a history of tumour evolution may, for instance, be constructed to investigate functional or adaptive differences between ancestral and derived clones (for example, the metastatic abilities between different clones). Notwithstanding, spatially separated biopsies may be necessary for sampling even greater genetic variability within the tumour.

Owing to limited sample size in patient number, the correlation between clinical phenotypes (for example, patient survival) and the ITH profiles are still provisional. In addition, the number of intra-hepatic metastases presented here is still small. Further studies with a larger cohort will be needed to further confirm and extend the conclusions of this study. The natural history of tumour progression is often described as belonging to either one of two classical models: either a linear evolution model with sequentially acquired driver mutations (late diversification)[46] or branched evolution with many standing lineages (early diversification)[47]. In our study, a wide variety of genetic sharing between tumour sectors (both early and late diversification) was observed[12]. Applying ITH analyses, the discovery of actionable mutations will probably provide hopeful new avenues for more effective targeted therapies for HCC. These data suggest that an understanding of ITH has the potential to illuminate mechanisms driving tumour evolution and guide therapeutics selection for personalized medicine.

## Methods

**Patient recruitment and spatial dissection.** All patients had surgical resection performed by the joint Hepato-Pancreato-Biliary Surgery service of the National Cancer Center Singapore and the Singapore General Hospital. Pre-operative diagnoses and staging were performed by multi-phasic radiological scans (computed tomography or magnetic resonance imaging) and diagnoses were confirmed histologically after surgery. All nine HCCs were early-stage tumours (AJCC stages 1 and 2). The study was approved by the Central Institution Review Board of SingHealth of which both National Cancer Center Singapore and SGH were constituent members (CIRB 2012/669/B) and each patient gave informed written consent. Entire resected surgical specimens were retrieved immediately after removal from the surgical field in the operating room and transported on ice in a temperature controlled cooler container to a pathologist at the SingHealth Tissue Repository situated in an adjacent building. Each specimen was then measured and photographed. Normal non-tumour liver tissue at least 2 cm away from the tumour was harvested and labelled as normal liver (control). A single slice

was then made in the tumour through the capsule and the cut section of the liver tumour photographed. The tumour was then inspected for necrosis, fibrosis, haemorrhage and cystic changes. Multiple pieces along the long axis of the tumour were then harvested, often at least 1 cm apart and cut into pieces measuring $1 \times 0.5 \times 0.5$ cm. Circumstances such as cystic change, haemorrhage or necrosis would reduce the number of tumour pieces harvested. The numbers of sectors harvested depend on the size of the tumours.

All harvested tissues were snap frozen using liquid nitrogen and stored afterwards in a $-80\,°C$ freezer. Matching pieces were collected for routine processing and histological examination. Real-time quantitative PCR (Roche COBAS Ampliprep/COBAS TaqMan HBV Test version 2.0) was used to quantify the viral load in the HBV positive patients.

**DNA sequencing and somatic variant calling.** DNA was extracted from the normal and tumour tissues independently using Qiagen DNeasy Blood and Tissue kit. After quality check with electrophoresis, DNA were sonicated to shorter fragments using the Covaris system. Following quality check with Agilent 2100 Bioanalyzer, DNA fragments were end-repaired, ligated with sequencing adapters, amplified before WGS or WES by the sequencing platform at the Genome Institute of Singapore. For WES, NimbleGen SeqCap EZ Human Exome Library v3.0 was used for exome capture before sequencing. Illumina sequencing platform was used for both WGS and WES (Earlier samples were done using WES and subsequent samples were processed using WGS).

Raw sequence reads were mapped to the human reference genome (hg19) with the Burrows–Wheeler Aligner[48]. After removing duplicated reads using PICARD (http://broadinstitute.github.io/picard/), sequence data then follow through base quality recalibration and realignment using the Genome Analysis Tool Kit[49]. Somatic variants were called comparing tumour against normal using the Mutect programme (Version 1.1.4)[50] (Supplementary Data 3). Using sequenom platform (a mass spectrometry-based technology), a small subset of somatic variants were experimentally validated (Supplementary Note 1). Copy number alterations were called using the Sequenza package[20]. The significantly perturbed gene segments (Genome Identification of Significant Targets in Cancer, a.k.a GISTIC result) were extracted from a previous study ($q < 0.1$ from Supplementary Table 10 from Totoki et al.[42]) as the potential driver CNV for HCCs[42]. The R programme Rcircos was used to plot the circos plot for the copy numbers. Tumour purity is estimated using a modified version of ASCAT[51] using sequencing data (https://github.com/cancerit/ascatNgs). To match the WES data to the WGS in comparing the ITH profiles (for example, Fig. 3a), we downsampled the WES data to match the coverage used in the WGS sequencing. The downsampled data sets follow the same pipeline as our WGS data.

**HCC data curation and driver genes.** Somatic mutation across many cancer types were downloaded from a previous study through http://www.tumorportal.org/ (Supplementary Note 2). HCC data sets were curated from several previous publications. In total, 1,185 data sets (198 from TCGA[52], 242 from ICGC/French[53], 514 from ICGC/Japanese[42] including part of a recent WGS work[54], 231 from a Korean study[55] and 88 from a HongKong/Chinese study[56]) were downloaded from the publications and consortium websites. Mutational frequencies of each gene were then compiled across all these cases. MutSigCV was used to identify the driver genes for HCC across all these cases[18].

To test whether HCC driver mutations tend to be of higher frequency than the background mutations, we used a $\chi^2$-test to compare the proportion of mutations whose frequency is higher than 0.15 in the two sets (HCC driver mutations and background mutations). For mutation signatures in patient 9, we first compiled the mutation pattern across all sites and then projected the mutation pattern as a combination of 30 known signatures using a linear model similar to previous studies (http://cancer.sanger.ac.uk/cosmic/signatures)[53].

**Missing variability and ranking the evolutionary trajectories.** The upward trajectories were drawn by randomly sampling a subset of sectors from the total collection of sectors for each patient. For example, if the number of sectors for a tumour is N, we will randomly sample $n$ ($n >= 1$ and $n < N$) sectors and count the total number of detected somatic mutation for that subsample. For each $n$, we subsampled multiple times and computed the mean number of detectable somatic changes at that $n$ (denoted as $X_n$). Using the mean value of detected somatic change at $n = 1$ (that is, $X_1$) as the baseline, we calculated the fold increase in detectable somatic variant at each n (that is, $X_n/X_1$). To rank the trajectories observed in Fig. 3a, the slope (that is, the value at $x = 2$, that is, randomly sample two sectors) was ranked across cases. For patient 2 and 3 (WES), we used the downsampled data sets to calculate these values (that is, matching the sequence coverage see above).

**Phylogeny reconstruction and IBD.** Using the presence and absence of individual somatic changes, we calculated the hamming distances between individual samples from each patient. With a distance matrix between all sectors, neighbour-joining algorithm implemented in PAUP[57] was employed to infer the evolutionary relationship between all the samples. To access the statistical confidence of the phylogenetic tree, a bootstrap procedure is implemented by sampling from the list

of mutations with replacement[26]. The consensus tree and the bootstrap values was computed using the sumtrees programme implemented in DendroPy[58].

Using the allele frequencies extracted from the sequencing output (mpileup output from SAMtools[59]), we calculated the Fst statistic between samples using the Weir and Cockerham method[27]. To adjust the tumour purities of the sectors in calculating the Fst value, we recalibrated the allele frequencies by resampling the same number of reads taking into account the differences in tumour purities (Supplementary Note 3). With the Fst values calculated between each pair of samples, we plotted the relationship between physical distance of the sectors and the Fst values as a boxplot (IBD pattern; Fig. 3b and Supplementary Fig. 6). To fit a linear model between Fst and physical distances, we pooled all Fst values across all sites and samples (for each sample pair, we will calculate a Fst value for each somatic mutation). We then used all these information to fit a linear model using lm() in R. In the boxplot, the box marks the 1st and 3rd quantile of the distribution. The median is labeled with a band in the box. Upper whisker = min(max(x), Q3 + 1.5 *IQR) and lower whisker = max(min(x), Q1–1.5 *IQR). Q1 and Q3 are the first and third quantile of the distribution and IQR = Q3–Q1.

**The spatial model.** To simulate a spatial population which expands outwards, we first allocated a grid of points in the computer memory. A single cell was then seeded in the centre of the grid. The population expansion was conducted by allowing cells to divide in all six possible directions if the space is available. In each cell division, two descendent cells will be randomly allocated to the original location of the parental cell and the available nearby position. When cells divide, there will be x number of mutations accumulated. We assumed a mutation rate of 0.5 per genome per division. A series of cell populations were then sampled from different geographic locations and mutations from these cells were harvested for subsequent phylogenetic and IBD analysis. Many different versions of the simulations (for example, different population size, birth/death rates and selective schemes) were also attempted (Supplementary Note 4).

**HBV integration and integration hotspots.** Following an earlier publication on HBV integration, we identify all read pairs that can align to both human and the HBV genome using both BLAST and short reads aligner Batmis[60] (reads near to the HBV integration sites are more difficult to align with short reads aligner). Among the identified reads, the programme clusters the reads into groups based on their positions found in the human genome. Among each group of clustered reads, integration junction point can be identified among a subset of them (not all junction points can be found). The final result of the analysis is a list of integration sites together with the number of supported reads.

We downloaded the HBV integration (a total of 1,027 integration sites) found in two previous studies[41,42]. A slide window approach is conducted to scan the genome for regions of intense integration (defined as the integration hotspot). We used a slide window of 20 kb and a step size of 10 kb. Windows with the number of integration sites > 3 is designated as a hotspot region (a total of 24 windows or 0.016% of the genome; Supplementary Note 5).

**Drug targets and their genes.** A list of drugs and their associated targets were curated from the FDA website (http://www.fda.gov/drugs/scienceresearch/researchareas/pharmacogenetics/ucm083378.htm) and from http://www.genome.jp/kegg/drug/br08341.html.

**Data availability.** The raw data (WGS and WES) have been deposited in European Nucleotide Archive under accession code EGAS00001001603. The authors declare that all data supporting the findings of this study are available within the article and its Supplementary Information files or from the corresponding author upon reasonable request.

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

## Acknowledgements

We thank Huck Hui Ng, Axel Hillmer, Anders Skanderup and Tam Wai Leong for helpful discussions and comments. We also thank Sin Chi Chew, Rachel Choi-Hui Yi, Lynette Lai-Soh Han and the sequencing platform at the Genome Institute of Singapore for their help and support. This work is supported in part by National Medical Research Council (NMRC, Singapore) grant TCR15Jun006.

## Author contributions

P.K.-H.C., R.S.-Y.F. and W.Z. supervised the work. T.K.-H.L., S.-T.P., B.K.-P.G., A.Y.-F.C., S.P.C. and P.K.-H.C. collected the tissue samples. T.K.-H.L. performed the tissue dissection. Z.T., P.Y.N. and W.W.-J.S. conducted the genomic experiments for sequencing. M.H.N. curated the public data for HCC. J.Q.L., F.Y., Z.L. and C.C.K. contributed to the experimental validation of the SNVs. P.-Y.G., Y.J. and K.W.-K.S. investigated on the viral integration. W.Z., T.Z., M.-H.N., P.-Y.G. and K.W.-K.S. performed the genomic analysis. W.Z., T.K.-H.L., R.S.-Y.F. and P.K.-H.C. wrote the paper.

## Additional information

**Competing financial interests:** The authors declare no competing financial interests.

Supplementary Information

