## [Peer Review File · Nature Communications]

Reviewers' Comments:

Reviewer #1 (Remarks to the Author)

Authors conducted an analysis of intra-tumor heterogeneity (ITH) in hepatocellular carcinoma (HCC) by performing multi-regional sampling in 7 patients treated with resection (2 of them with intra-hepatic metastasis also analyzed). Authors performed WGS or WES and inferred evolutionary trajectories based on shared/private mutations. The study found higher variability in metastasis than anticipated suggesting rapid genetic diversification.

The issue of ITH is a very hot topic in oncology, but evidence of ITH in HCC is still quite scarce. This study is probably the most comprehensive analysis of ITH in HCC conducted so far, both in terms of number of samples and depth of the analysis. Regarding message, the study provides interesting views on HCC evolution and the impact of early/late diversification on tumor's aggressiveness. Overall, the study is well structured but there are a number of issues that should be addressed:

Major:

- It is unclear the association between early and late diversification and tumor aggressiveness. As per figure 4 (pie charts), I don't see clear differences in early relapse or micro-vascular invasion between both. It seems that AFP may be different, but with such low numbers these claims are too exploratory to have the reflection they have in the text. Authors should consider decreasing the strength of some of these claims.
- Authors should provide variant allele fraction for the mutations called. For example, what is the % of reads with mutant CTNNB1 in the samples with mutations. This is categorized in $> < 15\%$ in the figure, but actual numbers could also be informative (this could be provided as supplementary). Also, consider changing the colors since green and dark green are difficult to differentiate in that figure.
- Indicate the nature of adjacent non-tumoral liver: cirrhosis, fibrosis (stage), etc...
- The section "Targetable mutations from ITH and the clinical phenotypes" is quite vague and doesn't contribute to the core message of the paper. I'd consider re-shaping it.

Minor:

- Provide more detailed explanation of some of the column names of supplementary tables (e.g., suppl. table 4)
- Intro: Epidemiological data is referred to liver cancer (including both HCC and biliary cancer). Correct for accuracy.
- There are some typos: Page 10, line 293 (alpha-protein should be alpha-feto-protein).

Reviewer #2 (Remarks to the Author)

Although the authors have addressed some of my comments from the previous round of reviews, I still have major concerns:

Line 151 onward: The authors claim "Across all cases, sectors from one end of the tumor consistently grouped together..."

This is not supported by Figure 2A and I had raised similar concerns when I first reviewed the manuscript. Regions for patient 1, 4, 5 and 6 appear to branch off at the same point from the trunk and do not show any grouping. And in patient 7, samples from opposite ends of the tumour DO cluster together, contradicting what the authors claim in the text. This significantly reduces the credibility of their claims that growth of these tumours follows a isolation by distance pattern.

Line 168: The authors use the Fixation index (F_{ST}) described in reference 20 to further support their conclusion of a population structure that resembles isolation by distance. It appears to me that the F_{ST} index has been developed for the analysis of sexually reproducing populations. Why this is applicable to asexually reproducing cancer cells needs to be clarified.

In their rebuttal letter, the authors replied to one of my previous comments:

"Regarding the technical issue of Exome/WGS, we have down sampled the exome data to match whole genome in terms of coverage. We recalculated the statistics for measuring the level of heterogeneity (Figure 2d) using the downsampled exomes. The slopes have changed slightly. We have now reorganized the presentation in this part"

Unfortunately, downsampling can not resolve the problem I highlighted. WGS will mainly identify mutations in non-exonic regions that are not thought to be under strong selection whereas exonic regions are – this will almost certainly introduce a strong bias. What the authors could have done is to analyze only exonic mutation calls in the WGS samples and compared that data to exome sequencing calls of the remaining cases.

Line 220: "If the metastatic clones were bona fide minor clones from within the primary tumor, we would have expected a much longer migratory branch (because many low frequency mutations specific to that lineage would be revealed if taking a very minor clone out of a larger tumor)."

I cannot follow this logic. Whether metastatic branches are longer or shorter is influenced, among other things, by the mutation load in the founder cell of the cancer, the timing of the bottlenecking event, selective sweeps in the primary tumour and the mets and by the mutation rate and spectrum in different tumour subclones. I am not sure how clone size in the primary should influence this.

Minor comments:

Line 115: Reference 13 seems wrong here

Line 143: "In general, the CNV profiles across the genome are quite similar across sectors, consistent with earlier observations that CNVs are early events in tumorigenesis 18."

The authors cite a paper specifically analysing CNV profiles in pancreatic cancer. I would argue that it remains unknown whether CNVs are generally early events. It may be the case in these HCCs and in pancreatic cancers.

Line 182: "The preponderance of truncal positions of these mutations suggests that HCC drivers are early events during HCC development (Figure 2c, P value=0.001) and this matches the observation across a large number of tumor types 16."

Similar to my comment in the first reviewing round: I would entirely agree with the statement that KNOWN HCC drivers are predominantly truncal based on this data. However, the authors fail to acknowledge that drivers which may be relevant during the subclonal differentiation process have not been studied systematically and may remain unknown. As the authors approach can only

assess whether known drivers are truncal, it is not possible to say whether the much more general statement "...HCC drivers are early events during HCC development..." is true or not. Reference 16 which the authors cite by the way clearly states that KNOWN drivers are predominantly truncal.

Reviewer #5 (Remarks to the Author)

The intratumor heterogeneity (ITH) has been characterized and reported for multiple cancer types, and it influences tumor development and clinical outcomes. It is clear that ITH is a major factor in hepatocellular carcinoma (HCC) and thus needs to be studied with advanced genome approaches. In this study, the authors performed whole-genome or whole-exome sequencing for multi-regional samples from seven surgically resected HCC tumors. As a reviewer of a previous version of this manuscript that was submitted to Nature Genetics, I also noticed that the authors also collected intra-hepatic metastatic tumors from two of those patients, and compared with their original lesions. The authors addressed most of my comments in a reasonable fashion. They also added much missed information, supplementary tables and figures in this revised documents. This revised version is more convincing, and many aspects of data presentation are appropriately revised. The new conclusions drawn from the analysis of metastatic samples further increase the scientific novelty of this paper.

Prior to the submission of this revised manuscript, a similar study has been published (Xue et al. Gastroenterology. 2016 Apr;150(4):998-1008). Since this paper was published in April, it seems to me that it is necessary for the authors to compare and contrast their results to the Xue et al paper.

One additional request is to take full advantage of the genomic alterations. SNVs, CNVs, Indels and virus integrations have been determined in this study, and these genetic alterations could occur concurrently or independently. In this current manuscript, the authors only used SNVs and Indels in the phylogenetic analysis and obtained an Isolation-By-Distance pattern of HCC development. It is desirable to find a way to input all types of somatic alterations to analyze the tumor evolutionary trajectory.

some minor comments:

1. The statement "most of the somatic mutations in the driver genes are in high allele frequencies (at least 15%)" needs to be quantified and statistically tested.
2. The authors state that the genetic geographic relationship in HCC tumors resembles that of human populations. It is not convincing, so please consider removing this.
3. The figure 3c and 3d appear to be rudimentary. Perhaps needs improvement.

In addition, Reviewer #3 raised several critical comments for its original manuscript, including sequencing depth, rationales and expressions. The authors have now provided detailed response to those comments and have made substantial revisions accordingly. They have improved their sequencing data and analysis methodologies. In addition, they removed some of the overly confident statements, and reduced the representation of areas that were less strong, such as the correlation between evolutionary trajectories and clinical phenotypes. Considering that the authors could not provide additional experimental validations, those comments from Reviewer #3 have been largely addressed in the current manuscript

Reviewer #1:

Reviewer 1 stated that our study is “the most comprehensive analysis of ITH in HCC conducted so far, both in terms of number of samples and depth of the analysis” and “the study is well structured and provides interesting views on HCC evolution”. We thank the reviewer for these positive and encouraging comments.

Major concerns:

Q1. It is unclear the association between early and late diversification and tumor aggressiveness. As per figure 4 (pie charts), I don't see clear differences in early relapse or micro-vascular invasion between both. It seems that AFP may be different, but with such low numbers these claims are too exploratory to have the reflection they have in the text. Authors should consider decreasing the strength of some of these claims.

Thank you for the suggestion. We agree with reviewer that these correlations are very tentative. We have taken the advice to reduce this part and reorganize the results. We have now moved the clinical phenotypes to earlier part of the manuscript (Figure 2). The correlation between ITH and clinical phenotype has been significantly reduced in the new version.

Q2. Authors should provide variant allele fraction for the mutations called. For example, what is the % of reads with mutant CTNNB1 in the samples with mutations. This is categorized in > < 15% in the figure, but actual numbers could also be informative (this could be provided as supplementary). Also, consider changing the colors since green and dark green are difficult to differentiate in that figure.

We have amended the allele frequency information to the supplementary table S9. In addition, we have changed the light/dark green to blue/red colors. The new figures will allow better visualization of the allele frequencies. We thank the reviewer for helping with our presentation.

Q3. Indicate the nature of adjacent non-tumoral liver: cirrhosis, fibrosis (stage), etc...

We have added this information to Supplementary Table S1.

Q4. The section "Targetable mutations from ITH and the clinical phenotypes" is quite vague and doesn't contribute to the core message of the paper. I'd consider re-shaping it.

We agree with the reviewer. As mentioned earlier in Q1 (see above), we have moved this section to earlier part of the manuscript and reduce the presentation in this part and have merged Figure 4 to Figure 2.

Minor comments:

Q5. Provide more detailed explanation of some of the column names of supplementary tables (e.g., suppl. Table 4)

This is now expanded in the table legends of Supplementary Tables (including Table S4).

Q6. Intro: Epidemiological data is referred to liver cancer (including both HCC and biliary cancer). Correct for accuracy.

We have corrected this in the latest version of our manuscript.

Q7. There are some typos: Page 10, line 293 (alpha-protein should be alpha-feto-protein).

This phrase is now corrected in the latest version of our manuscript.

Reviewer #2:

Major concerns:

Q1. Line 151 onward: The authors claim “Across all cases, sectors from one end of the tumor consistently grouped together...” This is not supported by Figure 2A and I had raised similar concerns when I first reviewed the manuscript. Regions for patient 1, 4, 5 and 6 appear to branch off at the same point from the trunk and do not show any grouping. And in patient 7, samples from opposite ends of the tumour DO cluster together, contradicting what the authors claim in the text. This significantly reduces the credibility of their claims that growth of these tumours follows a isolation by distance pattern.

We thank the reviewer for this rigorous review and agree that the internal branches of the phylogenetic tree from patient 1, 4, 6 are indeed very short compared to terminal/truncal branches. Rapid population growth tends to create tree topologies like these (commonly described as the star genealogy where internal branches are very short compared to external branches). In light of the suggestion from this reviewer, we took the advice to re-analyse the whole dataset (including new data that we have now added in, see below) and re-evaluated the phylogenetic relationship. Now, we used approaches from Statistical Phylogenetics (i.e. bootstrap procedure) to assess confidence in the tree topology, and used FST to calibrate the isolation by distance pattern (linear model). We found that, for late diversification cases (patients 1, 4), the phylogenetic trees are indeed less strongly supported, even though the isolation by distance pattern (FST vs physical distance) is still statistically significant for patients 1 (Figure 2d). For patient 6, although internal branches are short visually, the WGS data does provide enough of statistical support for resolving the phylogenetic relationship. We therefore agree with the reviewer's concern on the phylogenetic relationships for these patients.

Inspecting the phylogenetic tree for other seven patients (except patient 1 and 4), we found that the phylogenetic relationships can be grouped into two subtypes. For patients 6 and 7, we happened to sample the most ancestral/basal clone (e.g. T2 in patient 6, T3 in patient 7). These ancestral lineages are shorter and branch off at the basal position of the tree. This leads to the scenario where the basal clone will branch off first and then subsequently tumor sectors from one end will diverge from the sectors from the other end. In our computational simulations, we found that if we sample the sectors from the origin, we can also generate phylogenetic trees similar to patient 6 and 7. In this revision, we have added a section in the maintext to explain this point and we have also added a cartoon in Figure 2c to illustrate the two subtypes of phylogenetic trees (with/without basal lineages).

In light of the above observations and reviewer's suggestions, we have modified our arguments and restricted our conclusion only to a subset of the patients with enough of genetic divergence (i.e. except patients 1 and 4). In addition, we confined the conclusion to those patients with enough of sectors. Taking into account all these complexity (including two types of phylogenetic trees), we state explicitly in the abstract “a large proportion of HCC with appreciable genetic diversity display a clear Isolation-By-Distance (IBD) pattern...” and echoed this change at various

places across the manuscript, that the IBD pattern is strong only in a subset of patients with enough of genetic heterogeneity.

Lastly, during the time of the revision, we have sequenced two additional patients (i.e. 12 WGS additional data). Sectors from one end of the tumor also group together (e.g. (T1,T2,T3,T4) vs (T5, T6) for patient 8 and (T1, T2) vs (T3,T4) for patient 9). Therefore, the patterns revealed from these new patients continue to support our observation on the isolation by distance pattern. We added these two new cases to strengthen our conclusion.

We appreciate the scientific rigor that the reviewer is giving to help with this work. We hope the new statement we made for the Isolation-by-distance pattern is more appropriate and is reflecting the pattern we saw across patients.

Q2. Line 168: The authors use the Fixation index (FST) described in reference 20 to further support their conclusion of a population structure that resembles isolation by distance. It appears to me that the FST index has been developed for the analysis of sexually reproducing populations. Why this is applicable to asexually reproducing cancer cells needs to be clarified.

Thank you for this thoughtful point. We read into deeper details about the historical development of FST. FST was developed by Sewall Wright¹. Wright wanted to model population structure in breeds of livestock and in natural populations. It is indeed the case that FST was first developed using sexually reproducing systems like domesticated animals and plants as examples. Following Wright's seminal work, subsequent developments by Masatoshi Nei², C. Clark Cockerham³ and Bruce Weir⁴ have brought FST to an analysis of variance (ANOVA) framework and these new derivations relaxed the original assumptions architected on sexually reproducing organisms²⁻⁴. FST has been interpreted as the proportion of the total variance in allele frequency caused by allele frequency differences between populations. Under this framework, we believe (as did others as referenced²⁻⁴) that FST can be calculated without references to the underlying biological model.

Intuitively, the population structure captured by FST measures the segregation of alleles in different populations. Sexual reproduction (recombination) provides opportunities for different genetic loci to have different evolutionary histories. Since there is no recombination in asexual populations, we can intuitively think of genomes within asexually reproducing species/populations as a single genetic marker (like mtDNA or Y chromosome). Indeed, FST can still be applied in this situation; similar to the situation where FST can be applied to mtDNA and Y chromosome data as well. Following this reviewer's comments, we have added the above materials and expanded the discussion on FST in supplementary note S3.

Q3. In their rebuttal letter, the authors replied to one of my previous comments: "Regarding the technical issue of Exome/WGS, we have down sampled the exome data to match whole genome in terms of coverage. We recalculated the statistics for measuring the level of heterogeneity (Figure 2d) using the downsampled exomes. The slopes have changed slightly. We have now

reorganized the presentation in this part" Unfortunately, downsampling can not resolve the problem I highlighted. WGS will mainly identify mutations in non-exonic regions that are not thought to be under strong selection whereas exonic regions are – this will almost certainly introduce a strong bias. What the authors could have done is to analyze only exonic mutation calls in the WGS samples and compared that data to exome sequencing calls of the remaining cases.

We thank the reviewer for this important point. In the last revision, we applied downsampling to address the question on the technological differences between WGS and WES across the patient cases. We apologise that that we did not consider the genealogical difference between exome and WGS previously. In this revision, we have re-constructed the phylogenetic tree based on the exonic regions for patients with WGS data. We find that except for patients 1 and 8, all the other patients have identical exonic and WGS phylogenies.

The difference in patient 1 is mainly due to the fact that the phylogenetic tree is star-like and the whole genome tree itself is also not strongly supported (Q1 above). The topological difference between WGS and WES tree for patient 8 is very small (If we measure the phylogenetic distance between these two trees, for example, using the Robinson-Foulds distance⁵, the difference between these two trees is in fact really small). We agree with the reviewer that natural selection may contribute to this small difference in the tree topology. However, as discussed in the previous question, for asexual populations like cancer, even though selective pressure will be different between exome and whole genome, the complete linkage (i.e. the absence of recombination) will tend to carry all parts of the genome together. In Systematic Biology, there is a classical problem on gene tree differences between different parts of the genome due to recombination (e.g. in the context of the gene tree/species tree dis-concordance⁶). We are not completely sure whether there exists a similar problem in Cancer. Statistical fluctuations can also potentially contribute to the difference observed here. Fully addressing this question will require datasets beyond our current study. The observation from our current data suggests that exonic and whole genome phylogenies often agree with each other.

In light of the reviewer's comments, we have now included the phylogenetic inference based on WES in Supplementary Figure S5b and discussed this observation in Supplementary Note S3.

Q4. Line 220: "If the metastatic clones were bona fide minor clones from within the primary tumor, we would have expected a much longer migratory branch (because many low frequency mutations specific to that lineage would be revealed if taking a very minor clone out of a larger tumor)." I cannot follow this logic. Whether metastatic branches are longer or shorter is influenced, among other things, by the mutation load in the founder cell of the cancer, the timing of the bottlenecking event, selective sweeps in the primary tumour and the mets and by the mutation rate and spectrum in different tumour subclones. I am not sure how clone size in the primary should influence this.

Thank you for pointing out this gap in logic. In the new version of the manuscript, we have explained the reasoning in more detail and added a new figure panel (Figure 3b, also attached below) to clarify the logic.

The length of the migratory branch (the lineage linking the metastatic tumor to the primary tumor) is contributed by two factors. (a) the divergence between the migratory clone and the parental clone (see the figure below). (b) the further evolution at the distant location. If either of the two components is high, then the migratory branch will be long (see the cartoon illustration below).

The statement pointed out by the reviewer is pertaining to the point (a) above. The idea is that if the migratory clone is a very minor clone from the primary tumor, population bottleneck will be very strong in the metastatic process. In other words, the difference between the parental clone and migratory clone will be very large, simply because that minor clone will carry many private mutations specific to that group of cells (see figure below). Rephrasing this, minor clones will tend to create stronger population bottleneck and subsequently lead to higher divergence between parental clone and the migratory clone.

In both of the two metastatic cases, we found that the length of the migratory branch is very short. This suggests that the population bottleneck in the metastatic process is not very strong. In other words, migratory clone is not a very minor clone in the primary tumor and migration phenotype pre-existed in the primary tumor in relatively high frequency at the time of metastasis.

We have now rewritten the section around this part of the text and draw a new figure to illustrate this point (figure 3b).

Minor comments:

Q5. Line 115: Reference 13 seems wrong here

We have corrected the citation here. (We used an ASCAT like procedure AscAtNGS to perform the purity estimate, <https://github.com/cancerit/ascatNgs>). In the latest version, we have cited this link in our Materials and Methods (ascatNgs is a method based on ASCAT, but are tailored for sequencing type of data. There is no publication for this work at this moment).

Q6. Line 143: “In general, the CNV profiles across the genome are quite similar across sectors, consistent with earlier observations that CNVs are early events in tumorigenesis 18.”The authors cite a paper specifically analysing CNV profiles in pancreatic cancer. I would argue that it remains unknown whether CNVs are generally early events. It may be the case in these HCCs and in pancreatic cancers.

We apologize for the citation. In an earlier submission, we were imposed a limit for references. So, we used only one reference from a recent pancreatic cancer study, which we agree on its own is misleading.

The timing of the CNV events in the history of tumorigenesis has been a heavily studied topic. We have now cited a few more references including one from Bert Vogelstein where they found that chromosomal instability occurs very early in colorectal neoplasia. In addition, we also cite a recent HCC study (Xue et al 2016) which also found early origin of CNV in liver cancers.

Q7. Line 182: “The preponderance of truncal positions of these mutations suggests that HCC drivers are early events during HCC development (Figure 2c, Pvalue=0.001) and this matches the observation across a large number of tumor types 16.”Similar to my comment in the first reviewing round: I would entirely agree with the statement that KNOWN HCC drivers are predominantly truncal based on this data. However, the authors fail to acknowledge that drivers which may be relevant during the subclonal differentiation process have not been studied systematically and may remain unknown. As the authors approach can only assess whether known drivers are truncal, it is not possible to say whether the much more general statement“....HCC drivers are early events during HCC development....” is true or not. Reference 16 which the authors cite by the way clearly states that KNOWN drivers are predominantly truncal.

We agree with this and now emphasize the point on known HCC drivers. In light of this comments, we have moved the presentation on known HCC drivers (originally Figure 2 and Figure 4) to an earlier part and combined that with Figure 2a. We have emphasized that known HCC drivers are often early/truncal events in the tumorigenesis of HCC.

In summary, we have tried our best to address the questions as fully as we can. We hope this reviewer likes our latest revision of our manuscript and again we thank the reviewer for all his/her constructive comments and suggestions.

Reviewer #5:

Reviewer 5 thought that “this revised version is more convincing, and many aspects of data presentation are appropriately revised. The new conclusions drawn from the analysis of metastatic samples further increase the scientific novelty of this paper”. We appreciate the positive comments from the reviewer.

Q1. Prior to the submission of this revised manuscript, a similar study has been published (Xue et al. Gastroenterology. 2016 Apr;150(4):998-1008). Since this paper was published in April, it seems to me that it is necessary for the authors to compare and contrast their results to the Xue et al paper.

Thank you for pointing out this recent study to us. We have now read this work quite carefully. The authors focused on HBV positive HCC with a diverse variety of multiple lesions (MLs, e.g. portal vein or bill duct tumor thrombi together with multi-centric occurrence). So the disease cohort is slightly different from ours (ours are early stage HCC. Xue et al are late stage HCC). There is no multi-sectoring done for the lesions from Xue et al study, so, the approach is also slightly different from our current work (our work is focusing more on the intra-tumor heterogeneity and Xue et al study was focusing more on inter-tumor variability). However, there are several interesting links from Xue et al study to our work. a) The existence of both early/late origin of intra-hepatic metastasis. b) The early origin of CNV events in HCC. We also now added this reference in the introduction and connected our results to this study at various locations across results as well discuss this paper in our discussion comparing ITH between ours and Xue et al study.

Q2. One additional request is to take full advantage of the genomic alterations. SNVs, CNVs, Indels and virus integrations have been determined in this study, and these genetic alterations could occur concurrently or independently. In this current manuscript, the authors only used SNVs and Indels in the phylogenetic analysis and obtained an Isolation-By-Distance pattern of HCC development. It is desirable to find a way to input all types of somatic alterations to analyze the tumor evolutionary trajectory.

This is a good point. In recent years, there are several developments using copy number variations to construct phylogenies for the tumor sectors. We have tried a few of these approaches in several different settings (not limited to liver cancer) with whole genome data (e.g. MEDICC and a few others, a good review of recent developments can be found at Syst. Biol. 64(1):e1–e25, 2015, Cancer Evolution: Mathematical Models and Computational Inference). We had limited success with these methods (The models are often over-simplistic and they generally perform well on simulated data, but in real data the results are often not sensible possibly due to the continuum of CNV events, which violates the assumptions of the model).

How to integrate multi-layer omics information (in this case, CNV, mutations and indels) is a challenging question facing the community. Most importantly, how to properly assign “weights” to the different events (CNV vs SNV/indel) is a challenging question.

For the specific setting here, we started with a very intuitive approach. We first infer the genome wide copy number profiles (logR and BAF) using an ASCAT like procedure (<https://github.com/cancerit/ascatNgs>) for all tumor sectors independently. We then used a computational procedure based on penalized least squares minimization (Piecewise Constant Fits)⁷. This allowed us to jointly segment the copy number profiles across all sectors for each patient. Subsequently, we tabulated copy number variations across patient sectors taking into account the purity values of the sectors (slightly different cutoffs for each sectors depending on the tumor purity). With this binary presence and absence data, we combined mutation data (SNV and indel) together with the CNV information to build the phylogenetic tree using the neighbour joining algorithm. We found that, adding the CNV profiles are not changing the phylogeny inferred solely from the mutation data. This is because the number of CNV events is relatively small (less than 100) and most of them are truncal. We have now included this information and discuss this point in supplementary Note S3.

Minor comments:

Q3. The statement “most of the somatic mutations in the driver genes are in high allele frequencies (at least 15%)” needs to be quantified and statistically tested.

We compiled a two-by-two contingency table for the background sites as well as for the driver genes. The two factors in the contingency table are high/low frequency (cutoff at 0.15) and background mutations/driver mutations. The chi-square test is highly significant. The new results are included in the latest version.

Q4. The authors state that the genetic geographic relationship in HCC tumors resembles that of human populations. It is not convincing, so please consider removing this.

We have taken the reviewer’s suggestion and have removed this statement.

Q5. The figure 3c and 3d appear to be rudimentary. Perhaps needs improvement.

Thank you for this suggestion. We have moved Figure 3d to the supplementary material and modified Figure 3c accordingly.

In addition, Reviewer #3 raised several critical comments for its original manuscript, including sequencing depth, rationales and expressions. The authors have now provided detailed response to those comments and have made substantial revisions accordingly. They have improved their sequencing data and analysis methodologies. In addition, they removed some of the overly confident statements, and reduced the representation of areas that were less strong, such as the correlation between evolutionary trajectories and clinical

phenotypes. Considering that the authors could not provide additional experimental validations, those comments from Reviewer #3 have been largely addressed in the current manuscript.

We appreciate the efforts the reviewer has taken to evaluate the reply we had for the other reviewer's comments. We thank the reviewer for the positive comments for our revision.

References

- 1 Wright, S. The genetical structure of populations. *Annals of eugenics* **15**, 323-354 (1951).
- 2 Nei, M. F-statistics and analysis of gene diversity in subdivided populations. *Annals of human genetics* **41**, 225-233 (1977).
- 3 Cockerham, C. C. Analyses of gene frequencies. *Genetics* **74**, 679-700 (1973).
- 4 Weir, B. S. & Cockerham, C. C. Estimating F-statistics for the analysis of population structure. *evolution*, 1358-1370 (1984).
- 5 Robinson, D. F. & Foulds, L. R. Comparison of phylogenetic trees. *Mathematical biosciences* **53**, 131-147 (1981).
- 6 Pamilo, P. & Nei, M. Relationships between gene trees and species trees. *Molecular biology and evolution* **5**, 568-583 (1988).
- 7 Nilsen, G. *et al.* Copynumber: Efficient algorithms for single-and multi-track copy number segmentation. *BMC genomics* **13**, 591 (2012).

Reviewers' Comments:

Reviewer #1 (Remarks to the Author)

I don't have additional comments in relation to my previous input on this manuscript. I still believe that it is one of the most comprehensive analysis of ITH in HCC conducted so far. There are some inherent limitations of the paper (e.g., small sample size, lack of experimental validation of some of the claims related to cancer), but overall, I think it provides a solid landscape of the geographic distribution of genetic defects in HCC.

Reviewer #2 (Remarks to the Author)

Line 243: "For patient 1, the metastatic clone connected to the trunk of the primary phylogeny and the migration occurred before the genetic diversification of the primary tumor."

This conclusion cannot be made from a phylogenetic tree. The clone certainly diverged early, but the time of migration cannot be ascertained from the available data. The metastatic subclone may even still be present as a minor subclone within the primary tumour and this may have simply evaded detection due to very low allele frequency.

Line 275: "The fact that the migratory branch was very short in both of these intra-hepatic metastases led to two important conclusions: First, the population bottleneck was not very significant. In other words, the migratory clone was not a minor clone in the primary tumor and already existed in high frequency when metastasis occurred."

These conclusions are not convincing. Does a 'not very significant' population bottleneck mean that multiple subclones migrated? And the second statement (not a minor clone) is conditional on the definition of a minor clone. The authors have not identified the migratory clone within the primary, despite sequencing 7 and 10 subclones from these two tumours. The fact that they still did not pick up the migratory clones would suggest to me that it is a minor subclone.

Reviewer #5 (Remarks to the Author)

Dr Zhai and colleagues have revised their manuscript entitled "The spatial organization of intratumor heterogeneity and evolutionary trajectories of metastasis in hepatocellular carcinoma", and they have answered all of my concerns in point-by-point responses. They compared their results with a recently released paper which used the similar experimental and analytical measures as well as the object of hepatocellular carcinoma. The intra-tumor heterogeneity discussed here could be compared with heterogeneity of multiple lesions in HCC from a perspective of clonal inter-relationship, and provide essential clues in HCC development and clinical relevance. They also make effort to include CNV information to analyze the tumor evolutionary trajectory. Although they have not used virus integration data still, but I accept their conclusions that utilized SNV data by far. Additionally, the words and figures of this manuscript are improved substantially. So, in general, I'm satisfied with the author's effort of revisions and convinced their conclusions in this manuscript.

Reviewer 1 and 3 are satisfied with our revision and there were no further questions raised.

Reviewer #2

Q1. Line 243: “For patient 1, the metastatic clone connected to the trunk of the primary phylogeny and the migration occurred before the genetic diversification of the primary tumor.” This conclusion cannot be made from a phylogenetic tree. The clone certainly diverged early, but the time of migration cannot be ascertained from the available data. The metastatic subclone may even still be present as a minor subclone within the primary tumour and this may have simply evaded detection due to very low allele frequency.

We agree with the reviewer and have corrected the presentation. We changed the presentation from “the migration occurred before the genetic diversification of the primary tumor” to “the genetic divergence occurred before the genetic diversification of the primary tumor”.

Q2. Line 275: “The fact that the migratory branch was very short in both of these intra-hepatic metastases led to two important conclusions: First, the population bottleneck was not very significant. In other words, the migratory clone was not a minor clone in the primary tumor and already existed in high frequency when metastasis occurred.” These conclusions are not convincing. Does a ‘not very significant’ population bottleneck mean that multiple subclones migrated? And the second statement (not a minor clone) is conditional on the definition of a minor clone. The authors have not identified the migratory clone within the primary, despite sequencing 7 and 10 subclones from these two tumours. The fact that they still did not pick up the migratory clones would suggest to me that it is a minor subclone.

Thank you for the rigorous thought. In light of the reviewer’s comments, we have now removed this conclusion from the latest version of the manuscript.

We want to thank the reviewer for the careful thought and the great help with the presentation of this work.